# Humor Styles Predict Self-Reported Sarcasm Use in Interpersonal Communication

**DOI:** 10.3390/bs15070922

**Published:** 2025-07-08

**Authors:** Liberty McAuley, Melanie Glenwright

**Affiliations:** Department of Psychology, University of Manitoba, Winnipeg, MB R3T 2N2, Canada; mcauleyl@myumanitoba.ca

**Keywords:** sarcasm, verbal irony, humor styles, face-saving, politeness theory, relational humor, teasing, affiliative humor, aggressive humor, self-defeating humor

## Abstract

We investigated how participants’ humor styles impact their sarcasm use. English-speaking participants (N = 179) completed online self-report measures of humor styles and sarcasm use. We conducted linear regressions to test whether their humor style scores could predict their sarcasm use scores. Participants with higher affiliative humor scores reported a greater tendency to use sarcasm in general and to use face-saving sarcasm to protect the social images of the speaker and addressee. People use face-saving sarcasm to enhance their relationships, to tease others, and to self-deprecate. Surprisingly, participants who scored high on aggressive humor reported using face-saving sarcasm often. We suspect this occurred because the aggressive humor and the face-saving scales contain conceptually similar items. Participants with high aggressive humor scores also reported frequently using sarcasm to diffuse frustration. Participants who scored high on self-defeating humor reported often using both face-saving sarcasm and sarcasm to diffuse embarrassment. Given that face-saving sarcasm use was uniquely predicted by affiliative humor, aggressive humor, and self-defeating humor scores, we suggest that face-saving sarcasm use has utility for people with a wide range of humor styles. Our findings highlight how an individual’s humor style shapes their flexible use of sarcasm in interpersonal relationships.

## 1. Introduction

Picture this situation. You get a terrible haircut, and your coworker says in a sarcastic intonation, “Your hair looks amazing”. In this situation, your coworker is using a sarcastic criticism, opposed to a literal criticism, to address your unfortunate haircut. Sarcasm is defined as saying something very different from what you really mean, often with an intent to criticize ([24]). There are many different motives for using sarcasm such as to express humor, to criticize others, to relieve tensions, to promote social bonding, and to save face ([8]). In this situation, your coworker is using face-saving sarcasm. Politeness theory ([2]) posits that, when someone uses a sarcastic criticism with the intent to save face for the addressee, the speaker shows care and empathy for the addressee’s feelings. Accordingly, addressees rate sarcastic criticisms as more polite than literal criticisms ([11]). We were interested in learning about people who regularly use sarcasm and face-saving sarcasm by examining individual differences in humor styles, which are the relational and intrapersonal functions of humor in daily communication ([15]). We were also interested in examining whether politeness theory ([2]) could explain the relationships between humor styles and sarcasm use within the context of interpersonal communication. The primary purpose of the present study was to investigate whether an individual’s self-reported sarcasm use is linked to their tendency to employ four humor styles: affiliative, self-enhancing, aggressive, and self-defeating humor.

## 2. Humor Styles

The way an individual uses humor says a lot about a person, particularly how they feel about themselves and others. [15] ([15]) identified four humor styles which fit into two categories: humor that is self-focused (i.e., self-enhancing and self-defeating humor) and humor that is relationship-focused (i.e., affiliative and aggressive humor). These two categories contain positive humor styles (i.e., self-enhancing and affiliative humor) and negative humor styles (self-defeating and aggressive humor; [15]). According to a meta-analysis by [9] ([9]), positive humor styles are related to an improved emotional wellbeing, a heightened physiological wellbeing, better therapy outcomes, and positive personality traits. On the other hand, negative humor styles are associated with a decreased emotional wellbeing and are socially frowned upon ([9]). To provide a comprehensive picture of how each humor type functions, we will now examine each style closer.

Self-enhancing humor, often coined as coping humor, involves preserving a humorous point of view amid life challenges. [9] ([9]) argue that self-enhancing humor has the power to increase self-worth and enhance positive affect. Self-defeating humor, also known as self-deprecating humor, involves making jokes at the expense of oneself often to conceal one’s true feelings and earn acceptance from others ([15]). [9] ([9]) contend that self-defeating humor works to conceal insecurities in a socially appropriate way to evade challenging deeper emotional issues. On the other hand, affiliative humor involves the use of humor to enhance social relationships and diminish social tensions, while aggressive humor is used to demean others with a disregard for the other’s emotions ([15]). According to [15] ([15]), individuals who use affiliative humor are seen as more approachable and less intimidating. On the contrary, [18] ([18]) assert that individuals who use aggressive humor are lower in self-confidence and therefore more likely to use socially disapproved humor. With these humor subtypes in mind, we will now describe a complex and often misunderstood type of humor known as sarcasm.

## 3. Sarcasm Use

Sarcasm has a negative connotation because sarcastic speakers can be perceived as meaner than literal speakers ([3]) and as having the intention to mock ([13]) or hurt the addressee ([19]). On the contrary, a growing body of literature has highlighted positive aspects of sarcasm use. [21] ([21]) suggest that sarcasm can serve several beneficial interpersonal communication functions such as diffusing interpersonal tensions, managing adverse feelings, and strengthening social relationships. Similarly, cross-cultural research has shown that individuals rate maintaining social relationships as a central motivation for using sarcasm ([1]). [21] ([21]) also reason that sarcasm use can be linked to important interpersonal qualities such as perspective-taking skills and emotional competence. Additionally, they argue that sarcasm allows the speaker to convey negative feelings in a more constructive way by exhibiting greater impulse control and contributing to increased emotional intellect ([21]). While this literature highlights the social benefits of sarcasm, we must also consider individual differences in self-reported sarcasm production.

Based on a factor analysis, [10] ([10]) suggest four categories of sarcasm use: (1) general sarcasm, (2) face-saving sarcasm, (3) embarrassment diffusion, and (4) frustration diffusion. General sarcasm use (Factor 1) is characterized by an individual’s likelihood to use sarcasm in scenarios in which it is ordinarily used, such as with a best friend ([10]). Face-saving sarcasm use (Factor 2) involves producing sarcasm in socially risky situations, such as with a new acquaintance ([10]). According to [10] ([10]), people often use face-saving sarcasm in these risky situations because sarcasm serves as protective factor, and it can save a speaker’s face. Embarrassment diffusion sarcasm use (Factor 3) occurs in positive circumstances such as in instances of accomplishments (i.e., when you are telling your family about your promotion; [10]). [10] ([10]) suggested that individuals use this type of sarcasm to demonstrate humility. Lastly, frustration diffusion sarcasm use (Factor 4) happens in irritating scenarios (i.e., when you stub your toe in front of a colleague) because it serves to bring humor to annoying situations ([10]).

Research has shown that sarcastic criticisms are perceived as more polite than literal criticisms, and that the addressees of sarcastic insults rated speakers as less serious and less negative than speakers for literal insults ([11]). Furthermore, compared to addressees of literal insults, addressees of sarcastic insults expressed less defensiveness towards the speaker, and they had a reduced need for an apology from the speaker ([11]). [20] ([20]) argued that sarcasm is beneficial when used in supportive relationships as the strong bond and closeness eliminates any negative feelings that could arise from a sarcastic utterance. In the present study, we were interested in exploring ways in which self-reported sarcasm use might be conceptually linked with humor styles.

## 4. Politeness Theory

Politeness theory ([2]) assumes that every individual has a social persona, known as their ‘face’, which they constantly strive to preserve within their social interactions. Every member of society is susceptible to ‘losing their face’ and they navigate social interactions under the assumption of a reciprocal vulnerability of face between the speaker and the addressee ([2]). Certain types of communication such as expressing criticism, insulting someone, and teasing innately violate an individual’s face and are known as ‘face-threatening acts’ ([2]). An individual must use politeness strategies to mitigate these face-threatening acts ([2]). We reasoned that politeness theory ([2]) could deepen our understanding of the dynamic between sarcasm use and humor styles. In the present paper, we considered face-saving sarcasm as a politeness strategy where the speaker uses indirect language to soften the inherently face-threatening act of expressing criticism, thereby protecting the addressee’s face. We additionally argue that embarrassment diffusion sarcasm and frustration diffusion sarcasm both could be considered as politeness strategies. When an individual uses embarrassment diffusion sarcasm, they make sarcastic comments to downplay achievements and to avoid face-threatening outcomes like being perceived as egotistical ([10]). When an individual uses frustration diffusion sarcasm, they are using indirect language to express anger with an aggravating situation such as losing one’s keys ([10]). By choosing to use indirect language instead of direct expressions of anger or rude language such as swearing, the speaker appears more polite.

## 5. Face-Saving Sarcasm and Humor Styles

Speakers use face-saving sarcasm in socially risky situations where they are not sure how their sarcasm will be received by the addressee such as with a stranger, with a new coworker, or when complimenting someone ([10]). The speaker chooses to use sarcasm in these tricky situations because sarcasm’s indirectness can protect the speaker’s face ([10]). This corresponds with the Tinge Hypothesis, which states that, when a speaker uses sarcasm to criticize, the literal meaning of the criticism can positively tinge the addressee’s perception of the underlying negative meaning ([5]). Face-saving sarcasm can allow the speaker to appear less rude to the addressee, and the speaker is able to take less ownership of face-threatening perceptions ([11]). Thus, using sarcasm for face-saving purposes allows the speaker to remain in control and to express themselves in a socially desirable way ([12]).

Previous research has shown some commonalities between the use of face-saving sarcasm and certain humor styles. Self-enhancing humor, as well as face-saving sarcasm, can be used as emotional regulation tools and are related to increased perspective-taking skills ([8]; [15]). Individuals who use face-saving sarcasm take the addressee’s perspective into account when making sarcastic remarks which shows an ability to see others’ perspectives. Similarly, individuals who use self-enhancing humor to uphold a positive outlook when tackling life’s challenges demonstrate increased perspective-taking ([8]; [15]). Additionally, face-saving sarcasm and self-defeating humor share a similar goal of gaining validation from others ([2]; [15]). Individuals who use face-saving sarcasm make indirect comments to protect their face, which upholds the image they wish to project in social interactions, thereby gaining social validation ([2]). Similarly, individuals also use self-defeating humor to earn acceptance from others ([15]). Furthermore, face-saving sarcasm and affiliative humor both play a role in promoting bonding in interpersonal relationships ([5]; [15]). On the contrary, face-saving sarcasm and aggressive humor show stark dissimilarities. When an individual uses face-saving sarcasm, they show a care and concern for the addressee’s feelings by protecting the addressee’s face which is to the benefit of the speaker–addressee relationship ([5]). When an individual uses aggressive humor, on the other hand, they show a neglect for the other’s feelings which tends to lead to isolation and can negatively impact the speaker–addressee relationship ([15]).

## 6. Embarrassment Diffusion Sarcasm, Frustration Diffusion Sarcasm, and Humor Styles

Self-defeating humor involves making self-deprecating jokes, which enhances one’s relationship with others ([15]), and embarrassment diffusion sarcasm involves using sarcasm to dilute positive situations and to appear humble to others ([10]). Given this information, we predicted that there would be a relationship between self-defeating humor and embarrassment-diffusion sarcasm use as they both involve putting oneself down to enhance interpersonal relationships. When a person uses self-defeating humor, they mock themselves, which allows them to appear more relatable and, therefore, likeable. Similarly, when a person uses embarrassment diffusion sarcasm, they are deflecting attention away from themselves, which allows them to display humility and, by consequence, also appear more likeable.

We additionally predicted there would be a relationship between frustration diffusion sarcasm use and aggressive humor. As we previously discussed, aggressive humor can be used to manage irritating situations ([15]). Likewise, frustration diffusion sarcasm is used to make light of annoying circumstances ([10]). We would argue that the relationship between aggressive humor and frustration humor can be explained by their shared objective of indirectly expressing negative feelings. When a person uses aggressive humor, they express criticism framed as a joke. In a similar vein, when an individual uses frustration diffusion sarcasm, they use sarcastic comments to convey frustration in a covert way. Together, both humor types can be used to express negativity in a way that is more covert and socially acceptable than overt expression.

## 7. The Present Study

The aim of the present study was to investigate whether the four humor styles (affiliative, self-enhancing, aggressive, and self-defeating humor) could predict an individual’s self-reported sarcasm use as predicted. To assess these individual differences, participants completed online surveys measuring humor styles and sarcasm use. We predicted that individuals who have an increased tendency to use affiliative humor should use sarcasm, in general, more frequently. This prediction was based on the literature demonstrating that face-saving sarcasm has social bonding benefits, which aligns closely with affiliative humor’s ability to enhance social connections ([15]; [20]). Furthermore, we proposed that individuals who have an increased tendency to use affiliative humor should also use face-saving sarcasm more frequently. Politeness theory ([2]) posits that face-saving sarcasm serves to mitigate face-threatening acts and reduce interpersonal tensions, so we would also expect this sarcasm subtype to be related with affiliative humor, which promotes interpersonal harmony and alleviates face-threatening acts ([15]).

Next, we expected that individuals who use self-enhancing humor often should show a greater tendency to use face-saving sarcasm. This prediction was justified by reports that face-saving sarcasm and self-enhancing humor can both be used to exercise emotional control and promote greater perspective-taking ([8]; [15]). Although self-enhancing humor is an intrapersonal style of humor, self-enhancing humor and politeness strategies both involve increased perspective-taking ([2]; [15]). Politeness strategies such as face-saving sarcasm involve increased perspective-taking with regard to the addressee’s feelings within interpersonal interactions, while self-enhancing humor involves increased internal perspective-taking to uphold a positive mindset when navigating personal hardships ([8]; [15]). Furthermore, we predicted that individuals who use aggressive humor more often should use face-saving sarcasm less often. This prediction was based on observations that aggressive humor use does not show consideration for the addressee’s feelings, whereas face-saving sarcasm involves taking the addressee’s perspective and feelings into account ([8]; [15]). Furthermore, according to [2] ([2]) politeness theory, aggressive humor use does not require the perspective-taking and empathy involved in a politeness strategy like face-saving sarcasm.

Next, we expected that individuals who show an increased tendency to use self-defeating humor should report using face-saving sarcasm frequently because both self-defeating humor and face-saving sarcasm involve making humorous remarks to gain social validation from others ([2]; [15]). When an individual uses self-defeating humor, they express humor at the detriment of oneself to gain acceptance from others ([10]). Similarly, an individual uses face-saving sarcasm to appear more polite and likeable thereby increasing social validation ([2]). Together, both communication mechanisms can be used to enhance social bonds.

We additionally expected that individuals who reported to use embarrassment diffusion sarcasm more would use self-defeating humor more. This prediction was based on how both embarrassment diffusion sarcasm and self-defeating humor involve making humorous comments at the expense of oneself to deflect from the situation at hand and to increase social bonds. We reasoned that one who uses embarrassment diffusion sarcasm to diminish their achievements would also tend to make humorous comments to their own detriment within their interpersonal relationships, as seen in self-defeating humor.

Finally, we argued that individuals who reported to use frustration diffusion sarcasm more would show a greater tendency to use aggressive humor. This prediction stems from how frustration diffusion sarcasm and aggressive humor are both used to cope with negative feelings. When an individual uses frustration diffusion sarcasm, they make sarcastic comments to manage aggravation in an indirect way, and when an individual uses aggressive humor, they express negative emotions in an indirect way ([10]; [15]). Both frustration diffusion sarcasm and aggressive humor are ways to express negativity in a socially acceptable manner.

## 8. Methods

### 8.1. Participants

Initially 200 participants were recruited from introductory psychology classes at a university in a medium-sized Canadian city. We included participants between 18 and 25 years old, because both scales used were originally validated with young adult university students, and we wanted a comparable sample to test our predictions for relationships between items on the two scales. To be precise, the Humor Styles Questionnaire ([15]) was first validated with young adult university students, then further cross-validated with a larger sample that included middle-aged adults. A total of 21 participants were removed from the data analysis for the following reasons: 4 participants were removed for being older than 25 years of age, and 17 participants were removed for providing incomplete data. This resulted in a final sample size of 179 English-speaking participants, where 129 identified as female (72.1%), 45 identified as male (25.1%), 2 identified as transgender (1.1%), 2 preferred not to say (1.1%), and 1 identified as nonbinary (0.6%). The age distribution consisted of 145 participants aged 18–20 years old (81.0%) and 34 participants aged 21–25 years old (19.0%). Upon completion of the study, participants received a course credit. This procedure was approved by the University of Manitoba Research Ethics Board 1 (Protocol # HE2023-0230).

### 8.2. Measures

Participants completed surveys on the Qualtrics website (https://www.qualtrics.com) to assess humor styles and sarcasm use.

### 8.3. Humor Styles Questionnaire ([15])

To measure an individual’s humor styles, 32 items were assessed using a 7-point scale, ranging from 1 = totally disagree to 7 = totally agree. The survey included 8 items for each of the 4 humor types: affiliative humor, self-enhancing humor, aggressive humor, and self-defeating humor ([15]). Affiliative humor is used to promote bonding with others, thereby strengthening relationships and reducing interpersonal tensions (i.e., “I enjoy making people laugh”; [15]). Self-enhancing humor is used to avoid negative emotions and sustain a balanced view of the situation at hand (i.e., “My humorous outlook on life keeps me from getting overly upset or depressed about things”; [15]). Aggressive humor is used to enhance the self, often to the detriment of an other’s feelings (i.e., “When telling jokes or saying funny things, I am usually not very concerned about how other people are taking it”; [15]). Finally, self-defeating humor is used to amuse others, often at one’s own detriment (i.e., “I will often get carried away in putting myself down if it makes my family or friends laugh”; [15]).

Mean scores were calculated for each of the humor styles, and a total of 11 negative items were reverse-coded. Higher scores on self-enhancing and affiliative humor are said to be harmless, whereas higher scores on aggressive humor and self-defeating humor can be detrimental to one’s wellbeing ([15]). In the current sample, the internal consistency for each humor style were as follows: affiliative humor (α = 0.75), self-enhancing humor (α = 0.79), aggressive humor (α = 0.74), and self-defeating humor (α = 0.82). Additionally, within the current sample, all items of affiliative, self-enhancing, aggressive, and self-defeating humor styles were significantly correlated with their respective mean subscale scores, with a significance level of *p* < 0.001, indicating good content validity.

### 8.4. Sarcasm Self-Report Scale ([10])

To assess an individual’s likelihood to use sarcasm, participants completed the sarcasm self-report scale ([10]). Participants responded to 16 items which inquired about their likelihood of using sarcasm in certain situations and contexts. The 16 items were rated using a 7-point scale, ranging from 1 = not at all likely to 7 = extremely likely. Sarcasm use was divided into 4 subscales: general sarcasm, face-saving, embarrassment diffusion, and frustration diffusion ([10]). The general sarcasm subscale (Factor 1), consisting of 6 items, assessed sarcasm that is used in situations where it is commonly expected and an individual’s self-perceived sarcasm use (i.e., “Likelihood that you would use sarcasm when insulting” and “How sarcastic are you?”). The face-saving subscale (Factor 2) consisted of 3 items and included sarcasm use in risky situations (i.e., “Likelihood that you would use sarcasm with someone you just met”; [10]). The embarrassment diffusion subscale consisted of 3 items. Embarrassment diffusion sarcasm (Factor 3), as defined by [10] ([10]), is the use of sarcasm in a positive context, such as during success, to minimize positivity and diminish successes, thereby displaying humility (i.e., “Likelihood of using sarcasm when you got a promotion and are telling your family”). Lastly, the frustration diffusion scale (Factor 4) consisted of 3 items and involves the use of sarcasm to minimize frustration in irritating situations (i.e., “likelihood to use sarcasm when you made a mistake on an assignment”; [10]). In the current sample, the internal consistency for the sarcasm subscales was as follows: general (α = 0.83), face saving (α = 0.73), embarrassment diffusion (α = 0.74), and frustration diffusion (α = 0.67). For the present sample, all items of the general, face-saving, embarrassment diffusion, and frustration diffusion subscales were significantly correlated with their respective mean subscale scores, with a significance level of *p* < 0.001, suggesting good content validity.

### 8.5. Procedure

Before participants began the study, they were presented with an electronic consent form outlining general study information such as the constructs measured, the mechanisms of measurement used, and the implications of consenting to the study. To participate in the study, participants had to agree to the details outlined in the consent form. Participants completed the scales in the following order: the Sarcasm Self-report Scale, the Humor Styles Questionnaire, and then a demographic measure.

## 9. Results

Bivariate correlations between an individual’s sarcasm use scores and their scores on the four humor styles are shown in Table 1. As illustrated, the strongest relationship was between an individual’s general sarcasm use score (Factor 1) and their affiliative humor score, which affirms the prosocial and relationship enhancing benefits of sarcasm. Additionally, a noteworthy correlation was found between an individual’s embarrassment diffusion sarcasm score (Factor 3) and their aggressive humor score.

We performed a series of linear regressions on humor styles to predict each sarcasm use subscale (see Table 2). Consistent with our first prediction, we found that an individual’s affiliative humor style score was a significant predictor of general sarcasm use (Factor 1), *F*(1, 178) = 41.18, *p* < 0.001. This suggests that people with an affiliative humor style report that they frequently use general sarcasm in appropriate social situations.

Then we examined whether each of the four humor styles would uniquely predict face-saving sarcasm (Factor 2) scores according to our hypotheses. When we first tested for multicollinearity amongst predictors, the variance inflation factor for all four predictors was good (i.e., range = 1.1–1.3) and the tolerance was acceptable (i.e., 0.89 for each predictor). When all four humor styles were entered into a simultaneous multiple regression analysis, the model was significant, *F*(4, 175) = 12.29, *p* < 0.001. Affiliative humor uniquely predicted face-saving sarcasm use, β = 0.17, *p* < 0.025. This result supports our prediction that individuals who have an affiliative humor style would also use face-saving sarcasm often. However, self-enhancing humor did not significantly predict face-saving sarcasm use, β = 0.12, *p* = 0.091. Contrary to our prediction, people who use self-enhancing humor did not report a greater tendency to use face-saving sarcasm. Also contrary to our prediction, aggressive humor uniquely predicted face-saving sarcasm use, β = 0.26, *p* < 0.001. People who report having an aggressive humor style also report using face-saving sarcasm more often, whereas we predicted a negative association between these two variables. In support of our final prediction regarding face-saving sarcasm use, people who have a self-defeating humor style are more likely to report using face-saving sarcasm, β = 0.18, *p* < 0.025. This analysis produced mixed support for our predictions regarding face-saving sarcasm.

Next, we tested our prediction that people with a self-defeating humor style would report frequently using sarcasm to diffuse embarrassment (Factor 3). As predicted, self-defeating humor style scores significantly predicted embarrassment diffusion sarcasm use scores, *F*(1, 178) = 13.43, *p* < 0.001. Finally, we tested whether aggressive humor style scores could predict frustration diffusion sarcasm use (Factor 4) scores. Consistent with our prediction, people who report having an aggressive humor style also report frequently using sarcasm in frustrating situations, *F*(1, 178) = 8.10, *p* < 0.001.

## 10. Discussion

Our results show that individuals who use affiliative humor style more frequently tended to use general sarcasm more often. As previously mentioned, general sarcasm use typically occurs in close interpersonal relationships (i.e., with a best friend). This interpersonal closeness allows for an openness to use sarcasm based on a mutual understanding and comfortability ([20]). Similarly, affiliative humor involves the use of humor between close individuals based on common ground to strengthen interpersonal connection ([15]). Furthermore, affiliative humor can be used within romantic relationships to reduce relational uncertainty ([17]). Therefore, it makes sense to assume that an individual who uses general sarcasm more often would have an affiliative humor style because both behaviors rely on the speaker and addressee having a shared perspective and a focus on nurturing relationships.

We also found that individuals who reported using affiliative humor more frequently tended to also report using face-saving sarcasm often. This corresponds with the literature indicating that face-saving sarcasm is a humorous way to criticize someone with minimal adverse effects on one’s social relationships ([16]). Similarly, when individuals use affiliative humor, they amuse others while preserving social bonds ([15]). Therefore, it would make sense that someone who uses face-saving sarcasm would also have an affiliative humor style due to their shared relational benefits. Moreover, if we interpret this relationship through the lens of politeness theory ([2]), both face-saving sarcasm and affiliative humor can be utilized as a strategy to soften criticism and protect the face of both the speaker and the addressee.

Our findings also suggest that individuals who use aggressive humor more frequently have an increased propensity to use face-saving sarcasm. Although this result was contrary to our predictions, we reasoned that this association could be due to how both face-saving sarcasm and aggressive humor are related to a need for control ([2]; [7]). Individuals who use face-saving sarcasm can control how they are perceived by the addressee to maintain their ideal social identity within their interpersonal relationships ([2]). In a similar vein, individuals who use aggressive humor exert control over their social interactions by using humor to assert dominance and superiority ([7]). Perhaps individuals who use face-saving sarcasm more often may also use aggressive humor more often due to their shared ability to exert control over their social interactions. However, the aggressive humor style scale contains items regarding sarcasm use, teasing, and using humor to criticize others, so this unexpected relationship may have occurred because the aggressive humor style scale and the face-saving sarcasm scale contain conceptually similar items.

Furthermore, our results showed that individuals who use self-defeating humor more often have an increased likelihood of using face-saving sarcasm. Speakers who use face-saving sarcasm tend to care about the addressee’s point of view, and they want to be perceived as more polite by the addressee ([2]). Similarly, speakers who employ self-defeating humor make humorous remarks at the expense of themselves to amuse others and thereby gain social acceptance ([15]). According to [4] ([4]), self-defeating humor can have beneficial effects on interpersonal relationships by helping to reduce hierarchy in instances of power imbalances. Along the same lines, we would argue that face-saving sarcasm can also influence power inequalities by allowing for the expression of criticism without direct confrontation. As a result, it seems reasonable to assume that individuals who use self-defeating humor more often would also use face-saving sarcasm more often, as they both are associated with a shared outcome of social validation and supporting relational harmony.

Our study suggests that individuals who use self-defeating humor more often also have an increased tendency to use embarrassment diffusion sarcasm. We believe this connection can be explained by a shared goal of gaining social approval. Individuals use self-defeating humor to put themselves down, often becoming the punchline of someone else’s jokes to fit in ([15]). Likewise, individuals will use embarrassment diffusion sarcasm to appear more modest when sharing information about their accomplishments ([10]). Self-defeating humor can bring people together to reclaim negative group stereotypes or labels, allowing the speaker to display humility ([6]). Both self-defeating humor and embarrassment diffusion sarcasm use serve as a mechanism to save face by helping individuals to appear less egotistical, humbler, and more relatable.

Lastly, our findings demonstrated that individuals that use aggressive humor more also tend to use frustration diffusion sarcasm more often. We reason that both aggressive humor and frustration diffusion sarcasm are used to cope with negative feelings ([10]; [15]). In using aggressive humor, an individual can convey criticism in an indirect way through the socially acceptable guise of a joke. By the same token, when an individual uses frustration diffusion sarcasm, they use sarcastic comments to vent their annoyances in a socially permissible way. When used with positive intentions, aggressive humor can bring people together and foster group cohesion ([4]). We suggest that frustration diffusion can also function to foster group cohesion when used to vent about shared negative experiences. Both methods of communication allow people to express negativity while avoiding explicit confrontation to maintain connections with others.

### 10.1. Limitations

We recognize the following limitations of the present study. First, our sample consisted of young adult university students, and this limits the generalizability of our results. Another limitation is that the two measures were presented online to all participants in the same order, which creates the possibility of an order effect. We also need to acknowledge the limitations of the scales we used. Despite its widespread use, the Humor Styles Questionnaire (HSQ; [15]) has noted psychometric limitations. The wording of the items in affiliative humor style scale is problematic because all items are easy to endorse, resulting in low discrimination between people with average and high affiliative humor styles ([23]). The HSQ’s construct validity and criterion validity have also been criticized ([22]). There has been less published scrutiny of the Sarcasm Self-report Scale (SSS; [10]). Recently, [14] ([14]) administered the SSS to people living in Poland, Turkey, and Canada, in their native languages, and reported that the scale’s principal components remained valid cross-culturally. However, all self-report measures are subject to a social desirability bias where respondents may tend to overreport positive behaviors and underreport negative behaviors. Furthermore, all self-report scales are subject to measurement error, so it is necessary to validate that these scales indeed measure what they purport to measure in future research with observational and laboratory studies of sarcasm use and humor styles. Additionally, although we used humor styles to predict sarcasm use subscales in a correlational design, the actual direction of this causal relationship can only be determined in future research using an experimental design.

Our study uniquely contributes to the humor literature by highlighting the positive outcomes of sarcasm use and face-saving sarcasm use and by uncovering specific associated humor style tendencies. This specificity allows us to see the benefits of sarcasm and its positive role within our social interactions. Our research also highlights the flexibility of individuals who use face-saving sarcasm often. Individuals who use face-saving sarcasm more frequently report that they use aggressive humor, self-defeating humor, and affiliative humor more frequently, which shows their ability to adapt their humorous comments to the social situations and audiences at hand. These findings ultimately position the use of face-saving sarcasm and humor styles as a possible conflict resolution strategy. When an individual combines the use of face-saving sarcasm with different humor styles, they can alter their response to any situation at hand and conquer a range of interpersonal conflicts.

### 10.2. Directions for Future Research

Building on these insights, we suggest that future research explore personality variables to help understand the characteristics of people who use sarcasm frequently. A key area of focus could be the big five personality traits: openness to experience, conscientiousness, extraversion, agreeableness, and neuroticism. By measuring these traits, we can then make meaningful connections between an individual’s sarcasm use and their unique personality traits. These connections will ultimately provide insights into how our personality shapes our tendency to use sarcasm in general and face-saving sarcasm in particular. Finally, it would be beneficial to explore how culture plays a role in the use of face-saving sarcasm and the various humor styles. It would be interesting to compare people from individualist cultures to collectivist cultures, where humor is less direct and group harmony is more valued. We suspect that a strategy such as face-saving sarcasm might be important in maintaining interpersonal harmony.

## 11. Conclusions

Our findings showed that individuals who used affiliative humor more tended to use both general sarcasm and face-saving sarcasm more. Our results also indicated that individuals who use aggressive humor more tended to use both face-saving and frustration diffusion sarcasm more frequently. This highlights a crucial juxtaposition: affiliative humor and aggressive humor are related to an individual’s likelihood to use face-saving sarcasm, yet they accomplish opposing social outcomes; affiliative humor increases relational bonding, and aggressive humor allows one to criticize others. These findings illustrate the powerful role of face-saving sarcasm as an adaptable communication mechanism and provide support for [2] ([2]) theory of face-saving sarcasm being used as a politeness strategy to soften face-threatening acts. Furthermore, we found that individuals who use self-defeating humor more were also more likely to use face-saving sarcasm and embarrassment diffusion sarcasm. Collectively, these results emphasize the versatility of an individual’s humor style and how it can influence the many ways in which an individual uses sarcasm in their interpersonal relationships.

## Figures and Tables

**Table 1 behavsci-15-00922-t001:** Bivariate correlation coefficients for sarcasm subscales and humor styles.

	r	*p*	95% Confidence Intervals	
			Lower	Upper
General sarcasm (Factor 1)—Affiliative humor	0.436 *	<0.001	0.309	0.549
General sarcasm (Factor 1)—Self-enhancing humor	0.157	0.037	0.010	0.298
General sarcasm (Factor 1)—Aggressive humor	0.285 *	<0.001	0.144	0.415
General sarcasm (Factor 1)—Self-defeating humor	0.185	0.014	0.039	0.324
Face-saving sarcasm (Factor 2)—Affiliative humor	0.267 *	<0.001	0.124	0.399
Face-saving sarcasm (Factor 2)—Self-enhancing humor	0.244 *	<0.001	0.100	0.378
Face-saving sarcasm (Factor 2)—Aggressive humor	0.360 *	<0.001	0.224	0.482
Face-saving sarcasm (Factor 2)—Self-defeating humor	0.296 *	<0.001	0.156	0.425
Embarrassment diffusion sarcasm (Factor 3)—Affiliative humor	0.043	0.574	−0.106	0.189
Embarrassment diffusion sarcasm (Factor 3)—Self-enhancing humor	0.250 *	<0.001	0.107	0.383
Embarrassment diffusion sarcasm (Factor 3)—Aggressive humor	0.420 *	<0.001	0.291	0.534
Embarrassment diffusion sarcasm (Factor 3)—Self-defeating humor	0.262 *	<0.001	0.119	0.394
Frustration diffusion sarcasm (Factor 4)—Affiliative humor	0.153	0.042	0.006	0.294
Frustration diffusion sarcasm (Factor 4)—Self-enhancing humor	0.180	0.017	0.033	0.319
Frustration diffusion sarcasm (Factor 4)—Aggressive humor	0.210 *	0.005	0.065	0.347
Frustration diffusion sarcasm (Factor 4)—Self-defeating humor	0.267 *	<0.001	0.124	0.399

* *p* < 0.001.

**Table 2 behavsci-15-00922-t002:** Linear regression analyses on humor styles predicting each sarcasm use subscale.

Variable	B	SE	95% CI	β	*p*
General sarcasm use (Factor 1)					
Constant	7.12	3.22	[0.78, 13.47]	-	0.028
Affiliative humor style	0.47	0.074	[0.32, 0.63]	0.44	<0.001
Face-saving sarcasm use (Factor 2)					
Constant	−3.73	2.07	[−7.82, 0.36]	-	0.074
Affiliative humor style	0.10	0.05	[0.02, 0.19]	0.17	<0.025
Self-enhancing humor style	0.06	0.04	[−0.01, 0.14]	0.12	0.091
Aggressive humor style	0.14	0.04	[0.06, 0.21]	0.26	<0.001
Self-defeating humor style	0.09	0.03	[0.02, 0.15]	0.18	<0.025
Embarrassment diffusion sarcasm use (Factor 3)					
Constant	8.14	1.16	[5.84, 10.43]	-	<0.001
Self-defeating humor style	0.14	0.040	[0.07, 0.22]	0.27	<0.001
Frustration diffusion sarcasm use (Factor 4)					
Constant	8.66	1.29	[6.12, 11.21]	-	<0.001
Aggressive humor style	0.13	0.05	[0.04, 0.22]	0.21	<0.001

Note: N = 179.

## Data Availability

The dataset presented in this article is not readily available because participants were not asked for consent to share their data with other researchers.

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
