# Peer review of "Humor Styles Predict Self-Reported Sarcasm Use in Interpersonal Communication"

_behavsci, 2025, doi:10.3390/bs15070922_

Round 1
Reviewer 1 Report
Comments and Suggestions for Authors
In the essay titled “Humor styles predict self-reported sarcasm use in interpersonal communication,” the authors investigate how one’s sense of humor style impacts their self-reported tendency to use sarcasm in a range of social situations. More specifically, the authors indicate that this study is concerned with people’s use of sarcasm (general and face-saving) by examining individual differences in humor styles. Additionally, the authors draw on the politeness theory to explain why face-saving sarcasm use might be related to affiliative humor within the context of interpersonal communication.
The essay is well organized, clearly written, and engages relevant literature.
In my reading of this essay, the essay would benefit from a more elaborate discussion of two points. One is a more elaborate explanation of face-saving sarcasm. As it is, according to the authors, their chief interest in this study, it would be useful to comment in greater detail on the implications of face-saving sarcasm beyond the explanation that face-saving sarcasm use involves producing sarcasm in socially risky situations as protective factor. The second instance is a more elaborate explanation of the relationship between the aggressive humor style as a significant predictor of face-saving sarcasm. It seems to me that this dynamic ought to be explained more as per the results (Table 3) aggressive humor was a significant predictor of face-saving sarcasm use. Moreover, as aggressive humor is quite different from an affiliative humor style it would be useful to discuss the implications of such nuances for interpersonal relationships.
I hope the authors find these recommendations constructive and wish them the best of luck with their revisions.
Author Response
Reviewer 1
In the essay titled “Humor styles predict self-reported sarcasm use in interpersonal communication,” the authors investigate how one’s sense of humor style impacts their self-reported tendency to use sarcasm in a range of social situations. More specifically, the authors indicate that this study is concerned with people’s use of sarcasm (general and face-saving) by examining individual differences in humor styles. Additionally, the authors draw on the politeness theory to explain why face-saving sarcasm use might be related to affiliative humor within the context of interpersonal communication.
The essay is well organized, clearly written, and engages relevant literature.
In my reading of this essay, the essay would benefit from a more elaborate discussion of two points. One is a more elaborate explanation of face-saving sarcasm. As it is, according to the authors, their chief interest in this study, it would be useful to comment in greater detail on the implications of face-saving sarcasm beyond the explanation that face-saving sarcasm use involves producing sarcasm in socially risky situations as protective factor.
Thank you for this helpful suggestion. We have added the following text to page 4, lines 143-154: Speakers use face-saving sarcasm in socially risky situations where they are not sure how their sarcasm will be received by the addressee such as with a stranger, with a new coworker, or when complimenting someone (Ivanko et al, 2000). The speaker chooses to use sarcasm in these tricky situations because sarcasm’s indirectness can protect the speaker’s face (Ivanko et al. (2004). This corresponds with the Tinge Hypothesis which states that, when a speaker uses sarcasm to criticize, the literal meaning of the criticism can positively tinge the addressee’s perception of the underlying negative meaning (Dews & Winner, 1995). Face-saving sarcasm can allow the speaker to appear less rude to the addressee, and the speaker is able to take less ownership of face-threatening perceptions (Jorgensen, 1996). Thus, using sarcasm for face-saving purposes allows the speaker to remain in control, and to express themselves in a socially desirable way (Kalowski et al., 2021).”
The second instance is a more elaborate explanation of the relationship between the aggressive humor style as a significant predictor of face-saving sarcasm. It seems to me that this dynamic ought to be explained more as per the results (Table 3) aggressive humor was a significant predictor of face-saving sarcasm use. Moreover, as aggressive humor is quite different from an affiliative humor style it would be useful to discuss the implications of such nuances for interpersonal relationships.
The Discussion section now includes a more through explanation for the relationship between aggressive humor and face-saving sarcasm on page 11, lines 394-407.
I hope the authors find these recommendations constructive and wish them the best of luck with their revisions.
Thank you for your helpful feedback.
Reviewer 2 Report
Comments and Suggestions for Authors
I think that the topic is interesting, and I like that you drew on Brown & Levinson, but your article doesn’t give me enough useful information. Let me explain. I realize that you are constrained by the methodologies that you use and the current quantitative fashions, but for a sociolinguist like me with an ethnographic orientation, your article is very frustrating. I’m also somebody (a woman) with strongly negative attitudes toward what goes under the heading of “teasing,” and I don’t think that I use sarcasm in my friendships, or choose to hang out with people who do. I’ve always been repulsed by the idea that people would use sarcasm to show the strength of their friendship, and I tend to think of it as a very male thing.
I think that the main problem I have with the article is that there are no examples that I can use to get a handle on what you’re talking about. So, for instance on p. 3 the paragraph from lines 106-118, you describe a use of sarcasm and expect the reader to imagine examples. I can’t do that. This issue may be linked to your methodology that uses online surveys (and you provide us with no examples). I would expect, at the very least, to have an appendix to your article with a copy of the online survey. You say that there were “8 items” for each of the humor styles but then you give examples of how people might describe themselves in their use of a humor style. What do those descriptions have to do with the surveys?? And what was the sarcasm self-report scale with 16 items? Are you simply asking people what THEY think they do?? What about what they actually do—to say nothing of how their interlocutors interpret what they do?
Given this situation, all of your statistics are useless to me.
Also, after mentioning gender differences early in the article, you don’t appear to discuss your findings in light of the preponderance of women in your sample. Your findings seem to contradict what you had said earlier about tendencies for women. And what about generational differences? These were young people, born when?
The article also needs proof-reading and some stylistic adjustments for clarity.
The most interesting thing for me in the article was the discussion of “face-saving sarcasm” and perspective-taking skills. What I would really like to see is an ethnographically-oriented article in which you ask people to document what they do and then ask them to explain what they’re doing and how they are taking the perspectives of others. But then we really wouldn’t know if they had been successful unless we had the interpretations of their interlocutors, right?
Author Response
Reviewer 2
I think that the topic is interesting, and I like that you drew on Brown & Levinson, but your article doesn’t give me enough useful information. Let me explain. I realize that you are constrained by the methodologies that you use and the current quantitative fashions, but for a sociolinguist like me with an ethnographic orientation, your article is very frustrating. I’m also somebody (a woman) with strongly negative attitudes toward what goes under the heading of “teasing,” and I don’t think that I use sarcasm in my friendships, or choose to hang out with people who do. I’ve always been repulsed by the idea that people would use sarcasm to show the strength of their friendship, and I tend to think of it as a very male thing.
I think that the main problem I have with the article is that there are no examples that I can use to get a handle on what you’re talking about. So, for instance on p. 3 the paragraph from lines 106-118, you describe a use of sarcasm and expect the reader to imagine examples. I can’t do that. This issue may be linked to your methodology that uses online surveys (and you provide us with no examples). I would expect, at the very least, to have an appendix to your article with a copy of the online survey. You say that there were “8 items” for each of the humor styles but then you give examples of how people might describe themselves in their use of a humor style. What do those descriptions have to do with the surveys?? And what was the sarcasm self-report scale with 16 items? Are you simply asking people what THEY think they do?? What about what they actually do—to say nothing of how their interlocutors interpret what they do? Given this situation, all of your statistics are useless to me.
On Page 3 lines 94-107, we describe the Sarcasm Self-Report Scale, which was published by Ivanko et al. (2004). We have provided one example from each of the four factors from this self-report scale on page 3. The questions ask respondents if they would use sarcasm in hypothetical scenarios categorized according to a factor analysis. In response to this criticism, we have revised our description of this scale in the materials section to clarify that these examples refer to items on this scale on page 8, lines 303-306. Unfortunately, due to copyright law, and respect for the academic property that belongs to Ivanko et al. (2004), we cannot reproduce their scale in an appendix of the present paper.
Also, after mentioning gender differences early in the article, you don’t appear to discuss your findings in light of the preponderance of women in your sample. Your findings seem to contradict what you had said earlier about tendencies for women. And what about generational differences? These were young people, born when?
Given that gender differences are not a focus of this paper, we have removed the paragraph on gender differences from the Introduction.
The article also needs proof-reading and some stylistic adjustments for clarity.
We have thoroughly revised our manuscript for clarity and cohesion.
The most interesting thing for me in the article was the discussion of “face-saving sarcasm” and perspective-taking skills. What I would really like to see is an ethnographically-oriented article in which you ask people to document what they do and then ask them to explain what they’re doing and how they are taking the perspectives of others. But then we really wouldn’t know if they had been successful unless we had the interpretations of their interlocutors, right?
Your suggestion to examine the role of perspective-taking in naturalistic sarcasm production in future research is certainly interesting.
Reviewer 3 Report
Comments and Suggestions for Authors
This paper examined the relationship between sarcasm types and self-reported humor styles. The research was framed under Politeness theory and face-saving communication. Participants completed two measures and data were analyzed via series of hierarchical regression models. Results suggest that general sarcasm style was associated with an affiliative humor style, but also increased aggressive humor. Other sarcasm styles also were associated with various humor styles.
This is a very straightforward study based on one sample (N = 179) of college students who completed an online study with two measures. This left me concerned about the potential contribution and I wasn’t entirely persuaded by the authors’ conclusions. College students have their place in research, at the same time, limiting the sample to young college students greatly limits any generalizability of findings, and I find this to be a concerning shortcoming. To be clear, I don’t have a problem with the study structure or procedure, but replication(s) with different samples would go a long way in convincing an audience of the implications.
Additionally, participants were removed for being above an age threshold, yet this wasn’t explained beyond “outside the age cohort of interest.” But why? There was no mention in the Introduction or anywhere as to potential age effects and the politeness theory does not seem to be grounded in a younger adult perspective only. Without additional information, the age restriction presents another limitation in the potential generalizability or implications of findings.
I got a bit lost in the distinctions between the various types of sarcasm, and particularly in the distinction in analyses between general sarcasm and ‘face-saving’ sarcasm. I found it difficult to disentangle whether these were different strategies or if face-saving sarcasm was a subtype of general sarcasm. I see the description of the sarcasm self-report scale indicates that these are distinct, but in reading across the manuscript, it was hard to distinguish the difference. This may simply be due to the labels used in the measure itself, but I wonder if a different label would make the distinction easier for the reader to follow.
Were the measures administered in randomized order? If not, this is another potential limitation, and/or should be addressed.
More explanation about the rationale for using the stepwise hierarchal regression should be provided. It was unclear why the models were not conducted in the same fashion with the same predictors (e.g., general sarcasm prediction in Table 2 did not include self-defeating humor, yet prediction of face-saving sarcasm did include this variable.) If all models were similar, I would prefer to see the complete model results for each analysis. The distinction between * and ** in tables could be simplified to ** p < .01 to be in line with APA custom, as the distinction between .007 and .001 is fairly trivial in terms of the practical implications of results.
Author Response
Reviewer 3
This paper examined the relationship between sarcasm types and self-reported humor styles. The research was framed under Politeness theory and face-saving communication. Participants completed two measures and data were analyzed via series of hierarchical regression models. Results suggest that general sarcasm style was associated with an affiliative humor style but also increased aggressive humor. Other sarcasm styles also were associated with various humor styles.
This is a very straightforward study based on one sample (N = 179) of college students who completed an online study with two measures. This left me concerned about the potential contribution and I wasn’t entirely persuaded by the authors’ conclusions. College students have their place in research, at the same time, limiting the sample to young college students greatly limits any generalizability of findings, and I find this to be a concerning shortcoming. To be clear, I don’t have a problem with the study structure or procedure, but replication(s) with different samples would go a long way in convincing an audience of the implications.
Additionally, participants were removed for being above an age threshold, yet this wasn’t explained beyond “outside the age cohort of interest.” But why? There was no mention in the Introduction or anywhere as to potential age effects and the politeness theory does not seem to be grounded in a younger adult perspective only. Without additional information, the age restriction presents another limitation in the potential generalizability or implications of findings.
Thank you for this observation. On page 7, lines 260-265, we explain that “We included participants between 18 and 25 years old because both scales used were originally validated with young adult university students, and we wanted a comparable sample to test our predictions for relationships between the two scales. To be precise, the Humor Styles Questionnaire (Martin et al., 2003) was first validated with young adult university students then cross-validated with a larger sample that included middle aged adults.” We also recognize that is a limitation on page 12, line 444-445.
I got a bit lost in the distinctions between the various types of sarcasm, and particularly in the distinction in analyses between general sarcasm and ‘face-saving’ sarcasm. I found it difficult to disentangle whether these were different strategies or if face-saving sarcasm was a subtype of general sarcasm. I see the description of the sarcasm self-report scale indicates that these are distinct, but in reading across the manuscript, it was hard to distinguish the difference. This may simply be due to the labels used in the measure itself, but I wonder if a different label would make the distinction easier for the reader to follow.
On Page 3 lines 94-107, we describe that the Sarcasm Self-Report Scale was developed with a factor analysis, so each subscale represents a distinct factor and type of sarcasm. We have provided one example from each of the distinct four factors from this self-report scale. However, to better distinguish the sarcasm subscales conceptually, we have added factor numbers: general sarcasm (Factor 1), face-saving sarcasm (Factor 2), embarrassment-diffusion sarcasm (Factor 3), and frustration-diffusion sarcasm (Factor 4) throughout the manuscript.
Were the measures administered in randomized order? If not, this is another potential limitation, and/or should be addressed.
Participants completed the scales in the following order: The Sarcasm Self-Report Scale, the Humor Styles Questionnaire, and then a demographic measure. This detail has been added to the end of the Procedure section on page 8, lines 330-331. This limitation is acknowledged on page 12, lines 446-447.
More explanation about the rationale for using the stepwise hierarchal regression should be provided. It was unclear why the models were not conducted in the same fashion with the same predictors (e.g., general sarcasm prediction in Table 2 did not include self-defeating humor, yet prediction of face-saving sarcasm did include this variable.) If all models were similar, I would prefer to see the complete model results for each analysis. The distinction between * and ** in tables could be simplified to ** p < .01 to be in line with APA custom, as the distinction between .007 and .001 is fairly trivial in terms of the practical implications of results.
We have revised our results section to include a series of linear regressions (and one multiple regression) instead of stepwise hierarchical regressions. The variables entered into each regression were determined by our predictions stated in the introduction. For consistency, we have revised Table 1 so that all asterisks denote p <.001.
Reviewer 4 Report
Comments and Suggestions for Authors
This paper aims to explore how an individual’s sense of humor influences their self-reported tendency to use sarcasm across various social situations. However, the introduction lacks rigor, particularly in explaining why embarrassment-diffusion sarcasm and frustration-diffusion sarcasm were neither the focus nor included in the hypotheses, despite being important forms of sarcasm. Moreover, the use of stepwise hierarchical regression is questionable, as this method focuses only on statistically significant variables. Consequently, it does not allow for a clear understanding of the unique contribution of each humor style after controlling for other humor styles and individual factors, such as the Big Five personality traits. Additionally, the conclusion section discusses only face-saving sarcasm, neglecting the other three types of sarcasm that were analyzed in the Results section. This inconsistency weakens the overall coherence of the study. Without addressing these issues, the findings of the paper remain unconvincing.
Refer to the following comments:
Introduction:
p.3, line 119-120
“In the present study, we were particularly focused on general and face-saving sarcasm as they are the most widely applicable sarcasm subtypes.”
⇒Readers will not understand why the authors did not focus on embarrassment-diffusion sarcasm and frustration-diffusion sarcasm, as they do not have specific hypotheses regarding these two types of sarcasm.
p.5, hypotheses
“We proposed that individuals who have an increased tendency to use affiliative humor should use face-saving sarcasm more frequently.”
“we expected that individuals who use self-enhancing humor often should show a greater tendency to use face-saving sarcasm.”
“we predicted that individuals who use aggressive humor more often should use face-saving sarcasm less often.”
“we expected that individuals who show an increased tendency to use self-
defeating humor should report using face-saving sarcasm frequently because both self-
defeating humor and face-saving sarcasm involve making humorous remarks to gain social validation from others”
⇒Readers may not understand why the authors formulated hypotheses for only these two types of sarcasm.
Results:
The authors used stepwise hierarchical regression to examine the impact of humor styles on sarcasm. However, I believe this approach is not appropriate, as it focuses only on statistically significant variables. As a result, it does not allow us to assess the unique contribution of each humor style to sarcasm after controlling for the others. Moreover, stepwise regression is typically used for exploratory analysis, which is not suitable in this context since the authors had pre-specified hypotheses.
p.7, line 289-293
“As illustrated, the strongest relationship was between an individual’s general use sarcasm score and their affiliative humor score, which affirms the prosocial and relationship enhancing benefits of sarcasm. Additionally, noteworthy correlations were found between an individual’s face-saving sarcasm score and their affiliative and aggressive humor scores.”
⇒Readers may not understand why this result is inconsistent with your initial hypothesis that individuals who use face-saving sarcasm more frequently would use aggressive humor less.
Discussion:
p.12, line 415-416
“Our study suggests that individuals who use embarrassment diffusion more have an increased tendency to use aggressive humor.”
p.12, line 430-431
“These findings also demonstrated that individuals that use frustration diffusion sarcasm more frequently also tend to have a self-defeating humor style.”
⇒The authors did not propose specific hypotheses regarding the two types of sarcasm. Therefore, these discussions appear purely exploratory, making it difficult for readers to evaluate their validity.
p.12, line 454-456
“A key area of focus could be the big five personality traits: openness to experience, conscientiousness, extraversion, agreeableness, and neuroticism.”
⇒This study focuses on individual differences in humor and sarcasm. It is important to control for the Big Five personality traits, as they can significantly influence the results.
Conclusion:
p.470-472
“Our results highlight the multifaceted nature of face-saving sarcasm use and emphasize its powerful role as a confliction resolution strategy within interpersonal relationships.”
⇒Readers may not understand why the authors focused only on face-saving sarcasm in the conclusion. They should also address the other three types of sarcasm, as these were analyzed in the Results section.
Author Response
Reviewer 4
This paper aims to explore how an individual’s sense of humor influences their self-reported tendency to use sarcasm across various social situations. However, the introduction lacks rigor, particularly in explaining why embarrassment-diffusion sarcasm and frustration-diffusion sarcasm were neither the focus nor included in the hypotheses, despite being important forms of sarcasm. Moreover, the use of stepwise hierarchical regression is questionable, as this method focuses only on statistically significant variables. Consequently, it does not allow for a clear understanding of the unique contribution of each humor style after controlling for other humor styles and individual factors, such as the Big Five personality traits. Additionally, the conclusion section discusses only face-saving sarcasm, neglecting the other three types of sarcasm that were analyzed in the Results section. This inconsistency weakens the overall coherence of the study. Without addressing these issues, the findings of the paper remain unconvincing.
Thank you for these helpful suggestions. We have revised our manuscript to include predictions for embarrassment-diffusion sarcasm and frustration-diffusion sarcasm on page 6, lines 242-256. We have also revised our analyses to include linear regressions instead of hierarchical regressions on pages 9-10. We did not collect data regarding the Big Five Personality traits. Our discussion section now includes conclusions about embarrassment-diffusion sarcasm and frustration-diffusion sarcasm on page 11, lines 421-430. The conclusions now describe all sarcasm subtypes studied on page 13, lines 491-502. We appreciate this constructive feedback, and the resulting revised manuscript is significantly more coherent.
Refer to the following comments:
Introduction:
p.3, line 119-120
“In the present study, we were particularly focused on general and face-saving sarcasm as they are the most widely applicable sarcasm subtypes.”
⇒Readers will not understand why the authors did not focus on embarrassment-diffusion sarcasm and frustration-diffusion sarcasm, as they do not have specific hypotheses regarding these two types of sarcasm.
Thank you for this suggestion. We have added predictions for embarrassment-diffusion sarcasm to page 6, lines 242-248. We have added predictions for frustration-diffusion sarcasm to page 6, lines 249-256.
p.5, hypotheses
“We proposed that individuals who have an increased tendency to use affiliative humor should use face-saving sarcasm more frequently.”
“we expected that individuals who use self-enhancing humor often should show a greater tendency to use face-saving sarcasm.”
“we predicted that individuals who use aggressive humor more often should use face-saving sarcasm less often.”
“we expected that individuals who show an increased tendency to use self-defeating humor should report using face-saving sarcasm frequently because both self-defeating humor and face-saving sarcasm involve making humorous remarks to gain social validation from others”
⇒Readers may not understand why the authors formulated hypotheses for only these two types of sarcasm.
We have added predictions for embarrassment-diffusion sarcasm and frustration-diffusion sarcasm to page 6, lines 242-256.
Results:
The authors used stepwise hierarchical regression to examine the impact of humor styles on sarcasm. However, I believe this approach is not appropriate, as it focuses only on statistically significant variables. As a result, it does not allow us to assess the unique contribution of each humor style to sarcasm after controlling for the others. Moreover, stepwise regression is typically used for exploratory analysis, which is not suitable in this context since the authors had pre-specified hypotheses.
Thank you for this constructive suggestion. We have used a series of linear regressions and a multiple regression to test our predictions regarding each unique predictor (see pages 9-10).
p.7, line 289-293
“As illustrated, the strongest relationship was between an individual’s general use sarcasm score and their affiliative humor score, which affirms the prosocial and relationship enhancing benefits of sarcasm. Additionally, noteworthy correlations were found between an individual’s face-saving sarcasm score and their affiliative and aggressive humor scores.”
⇒Readers may not understand why this result is inconsistent with your initial hypothesis that individuals who use face-saving sarcasm more frequently would use aggressive humor less.
On page 8, lines 336-338, we have revised this sentence to be: “Additionally, a noteworthy correlation was found between an individual’s embarrassment diffusion sarcasm score (Factor 3) and their aggressive humor score.” We discuss the relationship between face-saving humor and aggressive humor later in the results section for enhanced coherency.
Discussion:
p.12, line 415-416: “Our study suggests that individuals who use embarrassment diffusion more have an increased tendency to use aggressive humor.”
p.12, line 430-431: “These findings also demonstrated that individuals that use frustration diffusion sarcasm more frequently also tend to have a self-defeating humor style.”
⇒The authors did not propose specific hypotheses regarding the two types of sarcasm. Therefore, these discussions appear purely exploratory, making it difficult for readers to evaluate their validity.
We have added specific hypotheses regarding embarrassment diffusion sarcasm and frustration-diffusion sarcasm to page 6, lines 242-256.
p.12, line 454-456: “A key area of focus could be the big five personality traits: openness to experience, conscientiousness, extraversion, agreeableness, and neuroticism.”
⇒This study focuses on individual differences in humor and sarcasm. It is important to control for the Big Five personality traits, as they can significantly influence the results.
We did not collect data on participants’ Big Five personality traits. This is a suggestion for future research. We have revised this sentence for clarity on page 12, lines 478-481.
Conclusion:
p.470-472: “Our results highlight the multifaceted nature of face-saving sarcasm use and emphasize its powerful role as a confliction resolution strategy within interpersonal relationships.”
⇒Readers may not understand why the authors focused only on face-saving sarcasm in the conclusion. They should also address the other three types of sarcasm, as these were analyzed in the Results section.
We have rewritten the conclusion to contain a description of our results regarding all four sarcasm subscales on page 13, lines 504-517.
Thank you for your constructive feedback.
Round 2
Reviewer 3 Report
Comments and Suggestions for Authors
This is a revised version of a previously submitted manuscript. I reviewed the revision with attention towards my previous comments and updates to the manuscript.
The authors were thoughtful in their responses and I appreciated their elaboration in several portions of their manuscript. I really appreciated the further explanation of sarcasm styles and identifying via Factor 1, Factor 2, etc. I found this to greatly enhance the readability and interpretation of the findings.
The re-analyses using regression models also enhances the manuscript; it was much easier to follow and interpret their findings.
The authors gave reasonable explanations about age restrictions and elaboration about potential limitations of their work were fair and sensible.
Reviewer 4 Report
Comments and Suggestions for Authors
Thank you for revising.